# X-ray radiation excited ultralong (>20,000 seconds) intrinsic phosphorescence in aluminum nitride single-crystal scintillators

Richeng Lin[1,4], Wei Zheng [1,4 ✉], Liang Chen[2,4], Yanming Zhu[1], MengXuan Xu[3], Xiaoping Ouyang[2] & Feng Huang [1 ✉]

Phosphorescence is a fascinating photoelectronic phenomenon usually observed in rare-earth-doped inorganic crystals and organic molecular crystals, owning great potential in optical information storage, color display and biological dosimetry. Here, we present an ultralong intrinsic phosphorescence (>20,000 seconds) in AlN single-crystal scintillator through X-ray excitation. We suggest that the long afterglow emission originates from the intra-band transition related to native nitrogen vacancy. Some excited states formed by absorbing X-ray photons cannot satisfy the parity difference between initial and final states required by transition selection rule, so they cannot return to the ground state directly through radiation transitions but through several phonon-assisted intra-band transitions slowly. During this process, a long-term broad-spectra phosphorescence emission is formed. Investigating the X-ray excited phosphorescence emission in the AlN is of great significance to understanding the mechanism of phosphorescence in inorganic materials, and to realizing the practical applications in high-energy ray dosimetry.

[1] State Key Laboratory of Optoelectronic Materials and Technologies, School of Materials, Sun Yat-sen University, 510275 Guangzhou, China. [2] State Key Laboratory of Intense Pulsed Radiation Simulation and Effect and Radiation Detection Research Center, Northwest Institute of Nuclear Technology, 710024 Xi'an, China. [3] School of Nuclear Science and Technology, Xi'an Jiaotong University, 710049 Xi'an, China. [4] These authors contributed equally: Richeng Lin, Wei Zheng, Liang Chen. ✉email: zhengw37@mail.sysu.edu.cn; huangfeng@mail.sysu.edu.cn

Special radiative transition pathways, such as agminated triplet-state exciton recombination in organic molecular crystals, make the long-living excited-state electrons realize radiative recombination for persistent phosphorescence, with important application prospects in color display, optical information storage, and bio-imaging[1–5]. Long-term phosphorescence is usually found in rare-earth-doped inorganic crystals and metal-free organic molecules, benefiting from its special configuration of electronic structure to achieve the phosphorescence mechanism[6–10]. However, current phosphorescence emission from extrinsic rare-earth element and organic component is challenged in duration[11–13]. Faced with these obstacles, the intrinsic luminescence originating from native defects or formed excitons in inorganic materials shows greater potential for the both stable and ultralong phosphorescence emission[14,15].

In an inorganic semiconductor, light emitting usually is obtained by direct radiative transition from excited states to ground states, which are typical band-edge transitions and strict correspondence to the transition selection rules[2,16,17]. There is another case where the excited state and ground state have the same parity, and then the direct transitions between them are forbidden. However, through phonon-assisted transition process, it is possible for the transitions to be achieved[17–21]. In these situations, the electronic transition is usually similar to that of triplet excited ($T_1$) excitons, which is expected to obtain the persistent phosphorescence.

Aluminum nitride (AlN) is well known as an inorganic semiconductor with an ultrawide bandgap of about 6.2 eV, and its properties of phosphorescence, photoluminescence (PL), and thermoluminescence (TL) have attracted much attention[22–29]. Here we present an ultraviolet (UV) ultralong intrinsic phosphorescence (UIP; >20,000 s) of AlN single crystals (SCs) under X-ray excitation. Theoretical investigations show that the electronic structure configuration of AlN with nitrogen vacancy ($V_N$) can make above persistent phosphorescence mechanism be satisfied. For the AlN SCs, the absorption of X-ray photons produces abundant high-energy unsteady excited electrons. Due to quantum selection rule, the excited electrons relax to ground state through phonon-assisted transitions and in-band multi-process transitions by emitting persistent phosphorescence[30]. All the above investigations of X-ray excited UIP in AlN SCs helps to understand the quantum mechanism of UIP in inorganic semiconductors and to further realize the practical applications in high-energy ray dosimetry[31].

## Results

**X-ray-excited phosphorescence in AlN SCs.** The AlN SCs used in all experiments are grown by a high-temperature (2200 °C) physical vapor transport (PVT) method[32]. Due to such high temperature, it is difficult to obtain high-quality defect-free crystals, and that is why they are always accompanied by various defects such as aluminum vacancies ($V_{Al}$), nitrogen vacancies ($V_N$), and oxygen substitutional impurity ($O_N$)[33–37]. AlN is a typical polar semiconductor, which makes the tendency of produced defects different along crystal orientation, or in other words, the required formation energies required by defects are different[38–40]. The N atoms are usually used as end-capping atoms along [001] orientation, which can produce $V_N$ defects on the crystal face along [001] orientation under high-temperature growth condition[33]. Therefore, the obtained AlN SCs usually show a pale-yellow color. The X-ray diffraction (XRD) pattern of AlN SCs (Fig. 1a) shows strong characteristic diffraction peak (002) with obvious accompanying kα 2 peak and narrow full width at half maximum (FWHM; ~0.03 degree, fitted with a Gaussian function), indicating that the crystal possesses

single-crystal quality along **c** orientation. The inset shows a photograph of the AlN SC with a large diameter of about 15 mm and yellowish color. Under 193 nm ArF laser excitation, the PL spectra of the AlN SC shows an obvious band edge emission at 208 nm (~5.96 eV, Supplementary Fig. 1a) at room temperature[22,32,40–43]. In Supplementary Fig. 1b, the Raman spectra shows vibration modes of phonons at 248 cm$^{-1}$ ($E_2^{(1)}$), 658 cm$^{-1}$ ($E_2^{(2)}$), and 912 cm$^{-1}$ ($E_1^{(LO)}$) with a 488-nm laser as excitation at room temperature. The FWHM of $E_2^{(2)}$ mode is 4.59 cm$^{-1}$, which demonstrates the high crystalline quality of the AlN SCs[44].

For inorganic materials, defects will also give rise to fluorescence and phosphorescence emission[45]. In one case, compared with the direct radiative transition in band to band, there is one more relaxation process to trap carriers added to the in-bandgap defect transition, which makes the time of defect-related fluorescence emission longer than that of band-to-band fluorescence emission in most time. In another case, when the excited electrons are in trapped state for a long time and they cannot be transited back to the ground state directly, then they will be stored. The electrons in the trapped state are excited by additional energy, and then recombine to emit phosphorescence. Such a mechanism can be described by a typical phosphorescence model; in displacement coordinates, ground state ($S_0$) electrons are excited into singlet excited state ($S_1$). The electrons in $S_1$ state also can transit to triplet excited state ($T_1$) by obtaining an intersystem crossing energy of $\Delta E$. The transitions from $S_1$ to $S_0$ usually emit fluorescence (Flour.) and thermal-active delay fluorescence, as well as the long-living phosphorescence (Phosph.) from $T_1$ to $S_0$ (see Fig. 1b).

In Fig. 1c–e, the PL spectra of AlN SC with turning on/off X-ray excitation show that the AlN SC has an obvious long-living UV phosphorescence. Under continuous X-ray excitation, the AlN SC shows two emission peaks located at UV (center wavelength about 352 nm) and yellow (607 nm) regions, respectively. When the X-ray source is turned off for 3600 and 7200 s, the crystal still shows obvious 24 and 15% of the original UV emission, respectively. The inset illustrations show optical photographs of the phosphorescence of AlN collected by a UV-sensitive intensified charge-coupled device (iCCD). In Fig. 1f, both the time evolution of UV and the yellow emission are well fitted with exponential decay function.

**Temperature effect of the phosphorescence.** In Fig. 2a, the time-dependent spectra show that the crystal still has obvious UV emission after 20,000 s when the X-ray source is turned off. The decay of yellow emission is faster than that of the UV emission. According to the PL spectra (Fig. 2b), the crystal has a distinct UV emission after 7200 s, while the yellow emission almost disappears. In Fig. 2c, both of the decay curve of UV and yellow phosphorescence can be well fitted by the bi-exponential decay function, from which a fast time constant ($\tau_{1Yellow} = 258.4 \pm 18.7$ s, $\tau_{1UV} = 636.4 \pm 19.8$ s) and a slow time constant ($\tau_{2Yellow} = 6919 \pm 2114.4$ s, $\tau_{2UV} = 6095.9 \pm 503.2$ s) can be extracted; the time constants indicate that the excited carriers are relaxed by two different radiation recombination paths by producing the phosphorescence. To quantitatively characterize the phosphorescence, a persistent time ($T_d$) is defined as the time interval during which the emitting intensity decreases from 100 to 10%. Calculation shows that the $T_d$ of UV and yellow phosphorescence in AlN SCs are 9360 and 4320 s, respectively, from which it can be obviously concluded that the UV phosphorescence shows a longer $T_d$ than that of yellow phosphorescence, indicating a more significant weight of slow radiation recombination component in UV phosphorescence.

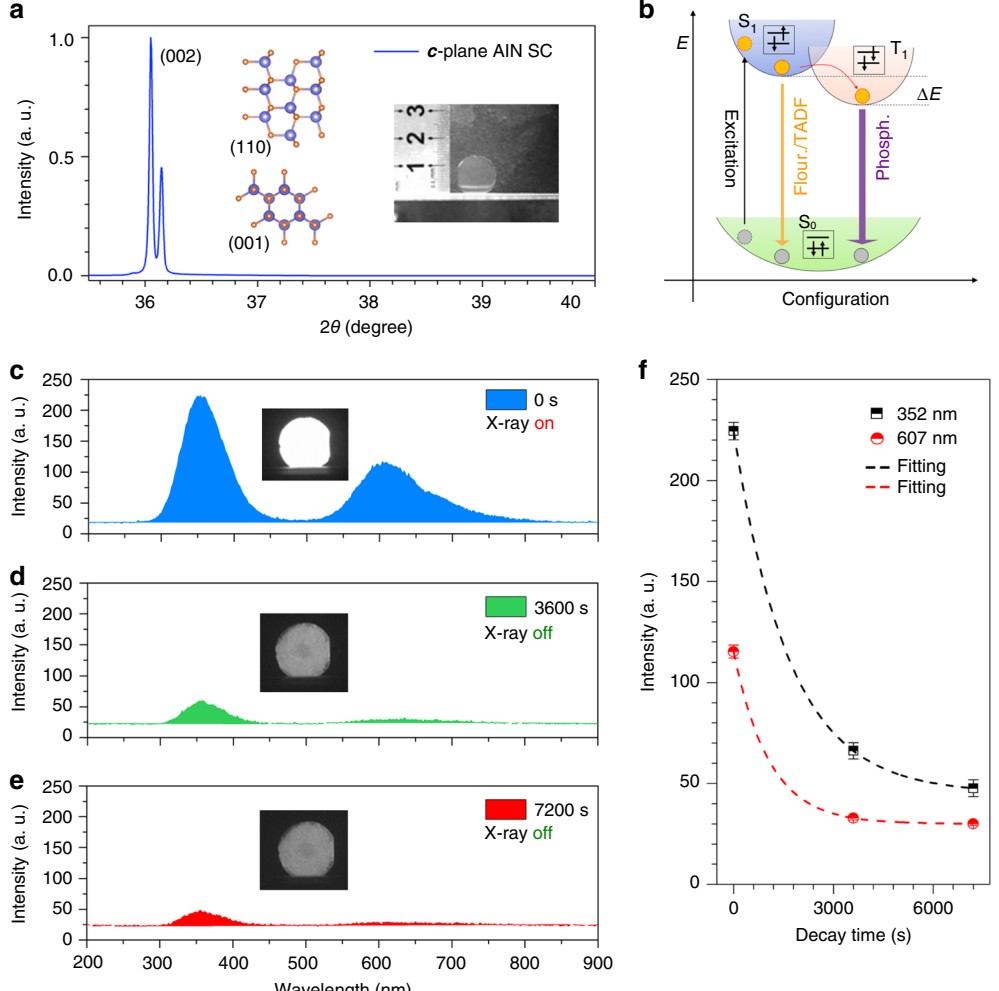

**Fig. 1 Crystal structure and phosphorescence of AlN SCs. a** XRD pattern of AlN SC shows obvious characteristic (002) peak (kα 2 peak). The insets show the crystal structure viewed on (110) and (001) faces; N atoms and Al atoms are in orange and blue, respectively. The optical photograph shows a sample of AlN SCs with the diameter of about 15 mm. **b** Displacement coordinate schematic of phosphorescence emission. Ground state ($S_0$) electrons are excited into singlet excited state ($S_1$) and the electrons in $S_1$ state transit to triplet excited state ($T_1$) by obtaining an intersystem crossing energy of $\Delta E$. The transitions from $S_1$ to $S_0$ usually emit fluorescence (Fluor.) and thermal-active delay fluorescence (TADF), and from $T_1$ to $S_0$ the transitions emit long-living phosphorescence (Phosph.). The phosphorescence spectra of AlN SCs after X-ray excitation for 0 s (**c**), 3600 s (**d**), and 7200 s (**e**) is compared. The insets show corresponding optical photographs collected by a UV-sensitive iCCD detector. **f** PL intensity plots with decay time show that the time evolution of UV and yellow phosphorescence matches well with exponential decay function.

Temperature-dependent PL spectra are used to further investigate the phosphorescence properties of AlN SCs. Under X-ray excitation, the crystals have a strong UV emission at low temperature (50 K), while the yellow emission is difficult to be observed (Fig. 2d). Besides, FWHM–temperature curve of the emission peaks (Fig. 2e) shows a reducing tendency with increasing temperature. Noticeably, different from conventional emissions from band-to-band transition and bound excitons, the emitting intensity in X-ray-excited AlN's phosphorescence is enhanced with increasing temperature (Fig. 2f). This temperature effect indicates that the emissions may be involved in the thermal-related (phonon-assisted) radiation transition. By investigating the phosphorescence, it can be found that, 2 min later after turning off X-ray excitation, the emission shows the same temperature-dependent effect (Fig. 2g–i), and the thermal-related radiation transition is also verified[46]. In temperature-dependent Raman spectra, a slight red shift and a broadened FWHM of the $E_2^{(2)}$ and $E_1^{(LO)}$ modes (Supplementary Fig. 2) are also observed, indicating a remarkable thermal-related lattice vibration in AlN SCs.

**Mechanism of the phosphorescence.** Under X-ray and 266 nm pulse laser excitation, the AlN SCs show strong UV and yellow emissions, which can be fitted to extract two components (Fig. 3a). As is known, AlN is a typical ultrawide bandgap SC with a bandgap of about 6.2 eV. Theoretically, the energy of excitation source should be greater than the bandgap, and thus the valence band electrons can be excited to conduction band effectively. The two emission also can be produced under 266 nm (~4.66 eV) laser excitation, indicating that there may be defect energy level in the bandgap. The role of defects also is investigated by density functional theory calculation introducing potential point defects (Supplementary Fig. 3), including $V_N$, $V_{Al}$, $O_N$, and couple of $V_{Al}$ and $O_N$[34,36]. In Fig. 3b, the $V_{Al}$ introduces an acceptor energy level near the valence band maximum (VBM), and the $V_N$ defect introduces two donor energy levels near the conduction band minimum (CBM) and an acceptor energy level near the VBM. Something worthy of attention is that the energy between two donor energy levels and acceptor energy level is in good agreement with the energy of UV (~3.52 eV) and yellow (~2.05 eV)

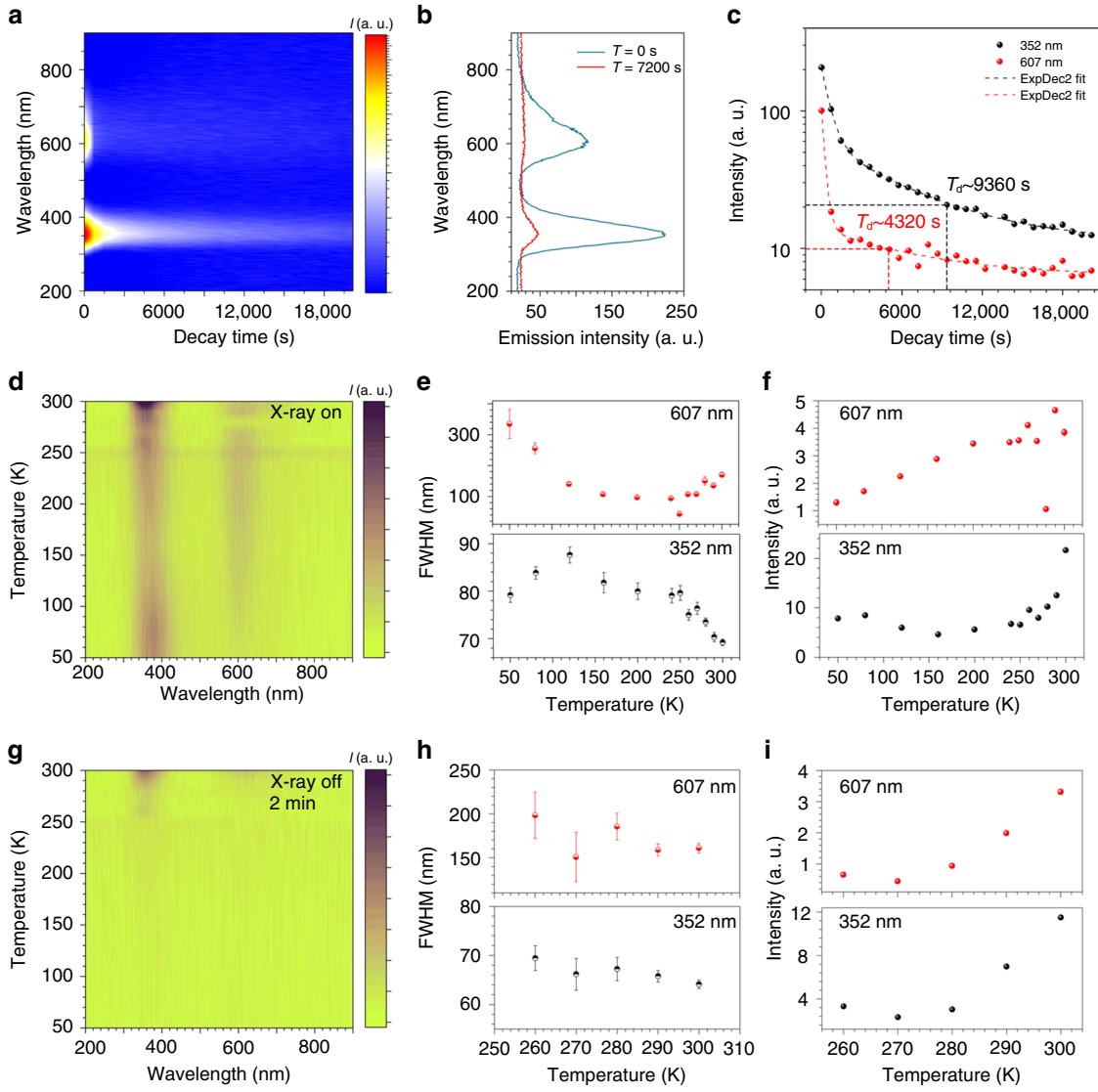

**Fig. 2 X-ray excited ultralong phosphorescence in AlN SCs. a** Time-dependent PL spectra of the AlN SC, showing an ultralong (>20,000 s) UV phosphorescence. **b** Steady-state PL spectra after turning off X-ray for 0 and 7200 s. **c** Decay curve of the UV and yellow phosphorescence, which are well fitted to bi-exponential decay function. The calculated $T_d$s of UV and yellow phosphorescence are 9360 and 4320 s, respectively. **d** Temperature-dependent PL spectra under X-ray excitation. **e, f** Extracted FWHM and intensities plotted with temperature, demonstrating that UV and yellow emission have thermal-related effect. **g–i** show similar temperature effect of the phosphorescence.

emissions, indicating that the emission can be attributed to the radiative transition from donors to acceptors.

In the energy band structure and partial density of states of AlN with $V_N$, N $2p$ orbits obviously contribute to valence band, conduction band, and even donor and acceptor energy level (Fig. 3c). Usually, the direct transitions between N $2p$ orbits will be forbidden because of their same parity. According to the quantum theory of optoelectronics, the parity of initial orbit and final orbit of electrons should be changed when the transition occurs. Therefore, the electron transition of $p$-to-$p$ orbits is usually forbidden, which stops the excited electrons on N $2p$ orbit from being directly transited to ground state on N $2p$ orbit. Considering the momentum selection rule, only the excited electrons those satisfy both momentum and parity conditions can realize direct transition.

Figure 3d shows the energy band diagram, which depicts the mechanism model of phosphorescence caused by native $V_N$ defects in AlN SCs. By absorbing high-energy X-ray photons, electrons in inner valence band are excited to the high-energy states of conduction band. A part of excited electrons can rapidly fluoresce by allowing direct radiative transitions, as such fast fluoresce exhibit strong self-absorption effect. There are still many unstable excited state electrons in N $2p$ orbit of the donor. The excited N $2p$ electrons can finally recombine with those holes trapped in N $2p$ + Al $3p$ orbit of the acceptor by absorbing a phonon or extra perturbation energy. Such recombination of $p$–$p$ orbital electron–hole pairs forming stable triplet excited state excitons due to orbital parity rule can produce an effective persistent phosphorescence[2,21]. For the AlN SCs, the presence of $V_N$ provides the electronic structure configuration that leads to the UV ultralong (352 nm, >20,000 s) phosphorescence.

**Defect and spectral analysis**. In the aspect of semiconductor fabrication techniques, annealing in source atmosphere is usually regarded as an effective method for reducing defects. Based on this method, high-temperature $N_2$ annealing is used to reduce the $V_N$ defects of AlN SCs on crystal surface. The PL spectra excited

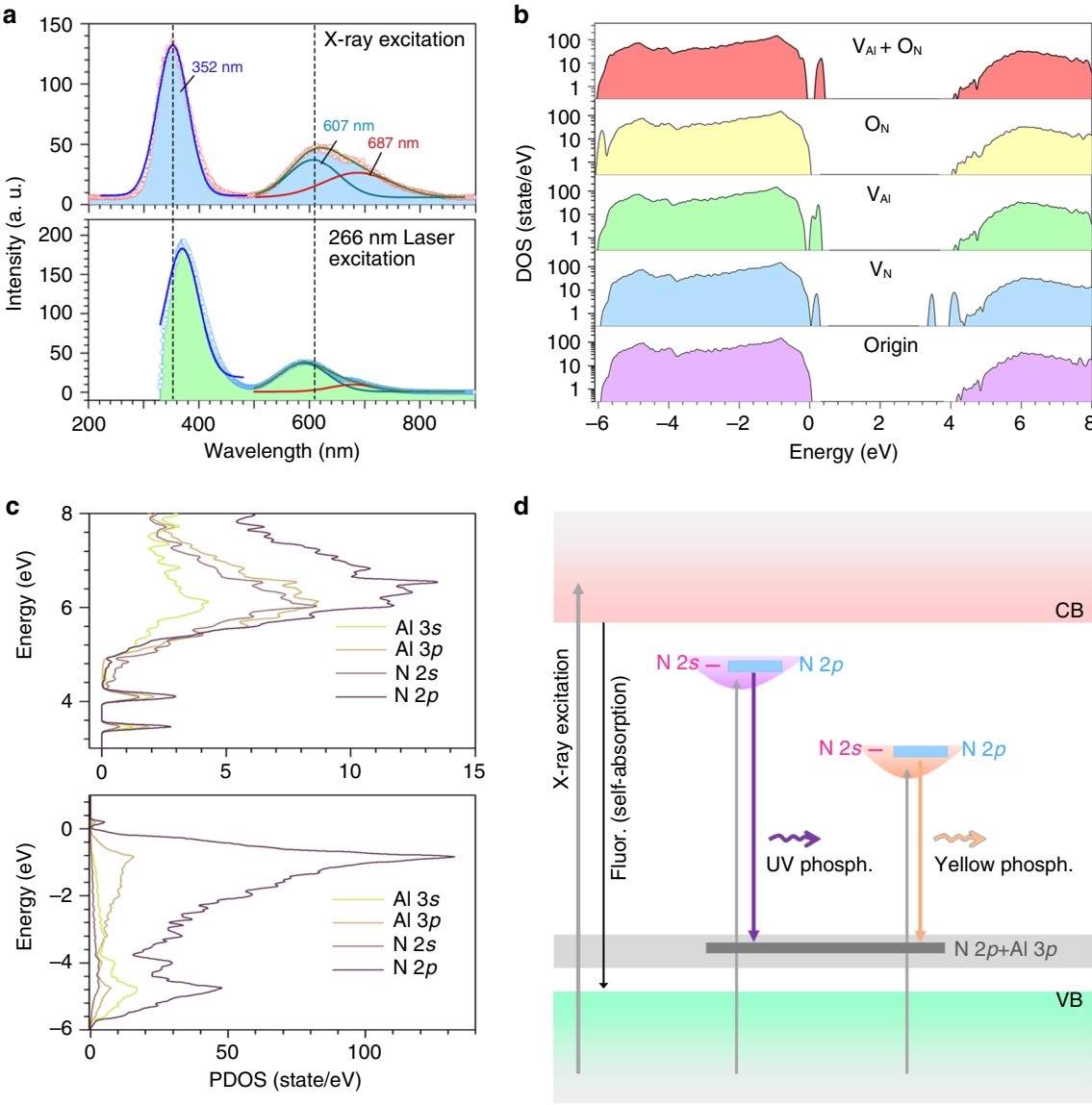

**Fig. 3 Mechanism of ultralong intrinsic phosphorescence. a** PL spectra of AlN SC under X-ray and 266 nm laser excitation, respectively. **b** The DOS of AlN with potential defects, including $V_N$, $V_{Al}$, $O_N$, and couple of $V_{Al}$ and $O_N$. **c** The PDOS of AlN with $V_N$ defect that introduces two donor energy levels and an acceptor energy level in inner bandgap. **d** Energy band diagram describes the radiative transition mechanism of phosphorescence (Phosph.) with stable transition from N $2p$ orbit to N $2p$ + Al $3p$ orbit.

by 266 nm laser after annealing shows an obvious decrease in the intensity of UV and yellow emission of the crystal (Fig. 4a). Under 325 nm excitation, the crystal only has yellow emission. In Fig. 4a, the PL spectra under different excitation present a reasonable variation that the UV emission disappears under 325 nm excitation, which is consistent with the calculation of $V_N$ defect energy level. Notably, the radiative transitions under low-energy (much less than X-rays) excitation are quite fast, which explains that the absorption transition strictly follows the quantum selection rule under low-energy excitation (see Supplementary Fig. 4). The excited-state electrons are allowed for direct radiative transition. After high-temperature $N_2$ annealing, the transmission spectra of AlN SCs show no obvious change, and the differential transmittance is shown in the inset (in Fig. 4b). Two variations of slope are presented in the spectra, indicating that the crystals should have two different edges of absorption. The energies of calculated absorption edge are also consistent with the results of the spectral and theoretical calculations above, corresponding to the two defect energy levels.

The analysis of elements and chemical bonds configuration after high-temperature surface treatment is performed by using an X-ray photoelectron spectroscopy (XPS), as shown in Supplementary Fig. 5. Generally, the photoelectronic spectra is strongly correlated with its chemical atomic state, so the XPS spectra is a useful tool for elemental analysis of crystal surfaces. In Fig. 4c, the elemental statistics of AlN SCs with high-temperature $N_2$ annealing show that the atomic proportion of $V_N$ is significantly reduced after $N_2$ annealing. For a detailed investigation, Al $2p$ and N $1s$ are observed, respectively (Fig. 4d). Obviously, the fitted characteristic Al–O–N and $V_N$ peaks of Al $2p$ and N $1s$ are reduced after $N_2$ annealing. The results of bond analysis after surface treatment further demonstrate that the luminescence mechanism is related to the $V_N$ defects in AlN SCs.

**Discussion**

In this work, we presented an ultralong (>20,000 s) intrinsic UV phosphorescence in AlN SCs under high-energy X-ray excitation. Unlike conventional rare-earth metal-doped inorganic

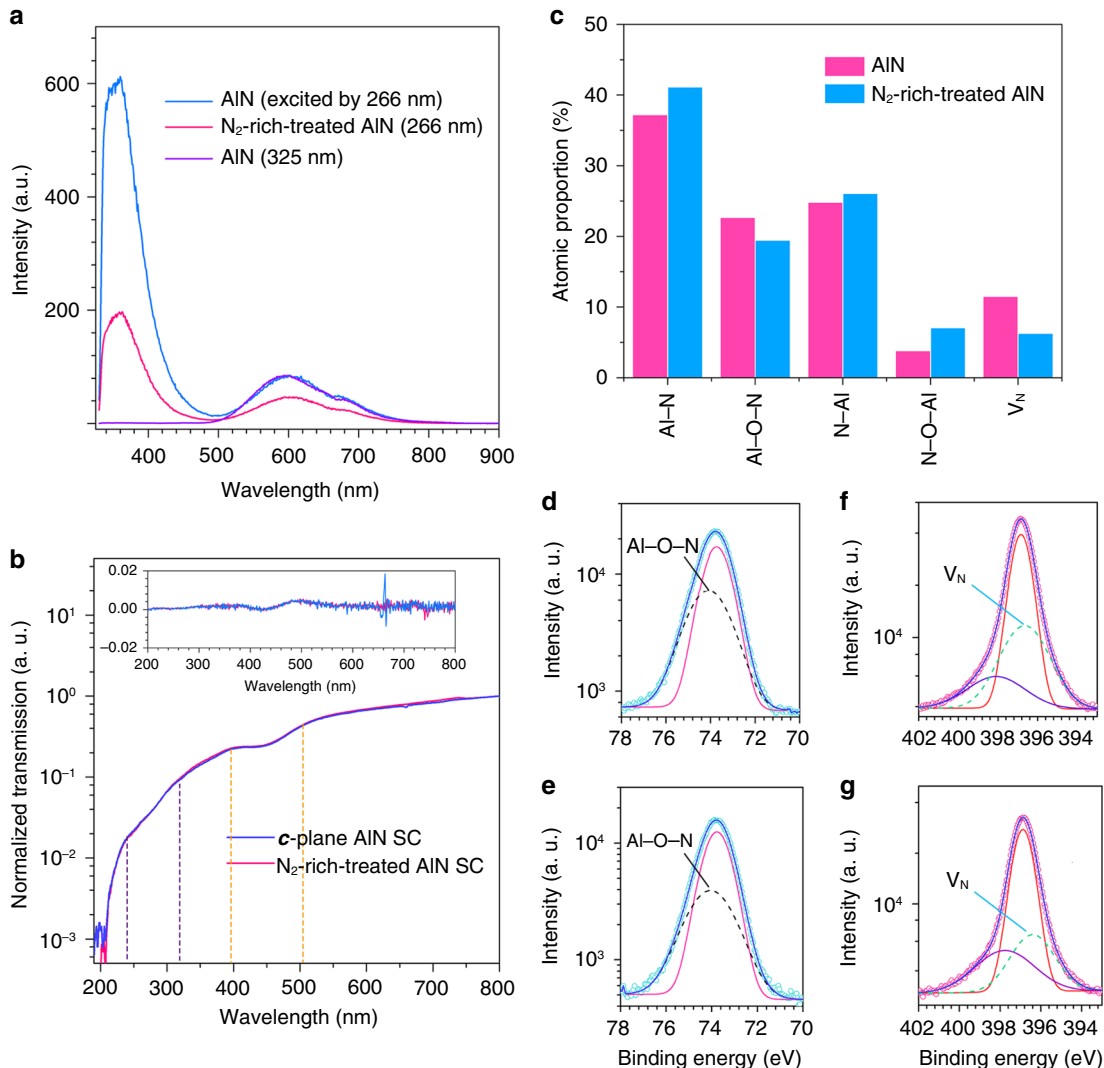

**Fig. 4 Analysis of surface defects in AlN SCs with high-temperature N$_2$-annealing process. a** PL spectra of the AlN SCs under 266 and 325 nm laser excitation. The N$_2$-rich-treated AlN shows obviously weaker PL emission than that of as-grown one under 266 nm laser excitation with equal excitation power. **b** Transmission spectra of the AlN SCs shows no obvious change after high-temperature N$_2$-annealing process. The inset is differential transmission (dT). There are two variations of slope in the transmission curve, corresponding to the absorption of V$_N$ defects. **c** Proportion of atomic binds in AlN SC and N$_2$-rich treated AlN SC. X-ray photoelectron spectroscopy (XPS) shows the characteristic peaks of Al 2p electrons (**d**, **e**) and N 1s electrons (**f**, **g**).

phosphorescent materials, the phosphorescence of AlN SCs originates from the native defects in crystal lattice. In photophysical observations, the phosphorescence emission involves multiple processes and distinct temperature effects. According to the theoretical calculations, it can be demonstrated that the UV and yellow emissions of AlN SCs are derived from V$_N$-defected transition. The parity-forbidden but phonon-assisted radiative transition between p-p orbits of V$_N$ donor and acceptor contributes to the transition mechanism of ultralong phosphorescence emission, which has been further investigated by analyzing the surface treatments. The discovery of ultralong phosphorescence in AlN SCs excited by X-ray is of great significance to understanding the intrinsic phosphorescence mechanism of inorganic materials and the applications of high-energy ray dosimetry.

## Methods

**Material preparation and crystal structure.** The AlN SCs were obtained by N$_2$-atmosphere PVT method in a tungsten crucible with the growth temperature of 2200 °C. The AlN crystals were cut to a wafer with the diameter of 15 mm and the thickness of about 500 μm (Fig. 1), and they were simply polished to obtain a smooth crystal surface for subsequent measurements. The XRD measurement used an Empyrean diffractometer with Cu target.

**Spectral measurements.** PL spectra of band-edge emission of AlN SCs was collected by an ocean optical spectra instrument and a 193 nm ArF laser. A commercial X-ray source was used as the excitation, set at a voltage of 30 kV and a current of 400 μA. An AlN SC was placed at the entrance of an ultraviolet–visible–near infrared spectrometer, and a highly sensitive photo-multiplier tube was used as photonic collector at the exit. PL spectra were collected when X-ray excitation was turned on. The phosphorescence spectra were collected each 120 s after the X-ray was turned off. The temperature-dependent PL spectra were collected by a vacuum-temperature control system with circulating liquid helium system. The AlN SC was clamped on the sample stage for spectral testing. The optical photograph of phosphorescence was obtained by an iCCD. The UV PL spectra was performed by using a 266 nm pulsed laser (2 MHz) and a 325 nm He–Cd laser. Transmission spectra were collected by using a Shimadzu UV-infrared spectrophotometer and the XPS by using Thermo Scientific's analysis system.

**Theoretical calculation.** The electronic structures of AlN with various point defects were calculated by Vienna Ab-initio Simulation Package. The defects were introduced by adding or removing corresponding atoms to/from supercell. The electron–core interaction was treated with projector augmented wave approximation. The Perdew–Burke–Ernzerhof generalized gradient approximation was

adopted to simplify the exchange correlation potential. The relaxation of each configuration was performed until the force on each atom was <0.005 eV/Å. A high-density $k$ grid of $13 \times 13 \times 13$ was used in self-consistent calculation.

## Data availability

The data that support the findings of this study are available from the corresponding author upon request.

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

## Acknowledgements

This work was supported by the National Natural Science Foundation of China (grant nos. 61604178, 91333207, U1505252, and 91833301) and the China Postdoctoral Science Foundation (no. 2018M643305).

## Author contributions

W.Z. and F.H. designed and directed this study. R.L., W.Z., and L.C. led the experimental work and paper preparation. Y.Z., M.X.X., and X.O. contributed to DFT simulations and experimental analysis.

## Competing interests

The authors declare no competing interests.
