## [Peer Review File · Nature Communications]

Reviewers' comments:

Reviewer #1 (Remarks to the Author):

The manuscript presents an interesting study of ultralong intrinsic phosphorescence (UIP) in aluminum nitride (AlN) single-crystal scintillators, and proposes a theoretical model which uses the thermal-assisted radiative recombination of parity-forbidden excited state (PFES) electrons. The 1-inch high-quality AlN single crystals notoriously difficult to produce at large sizes are first prepared by the authors. Interestingly, such AlN single crystals exhibit an UIP under X-ray radiation, and the PFES electron recombination is suggested to explain UIP mechanism, which is remarkable and important. Overall, I think that this manuscript merits publication in Nature Communication. Addressing the following minor issues will increase this paper's impact on the readership.

1) In Figure 2c, the UIP in AlN single crystal is remarkable, but the data and words in the inset are too small to read, and the fitting results are not detailed enough.

2) "In the AlN SCs, the presence of VN leads to the UV (352 nm) and yellow (607 nm) emission, and the ultralong (>20000 s) UV phosphorescence originates from the radiative transition of PFES electrons through thermal-assisted process, which is brought by the unique VN-related electronic structure of AlN." The authors' claim that the existence of intrinsic N vacancies in AlN is the major reason of UIP emission should be explained more clearly on its mechanism.

3) In Figure 3a, the PL spectroscopy of AlN both under X-ray and 266 nm laser can be fitted to three peaks, of which the characteristics of these fitting peaks should be given.

4) The language and grammar of this manuscript should be more specialized for readability, and more relevant citations should be added.

Reviewer #2 (Remarks to the Author):

The manuscript reports ultralong phosphorescence from AlN with a high concentration of N vacancies. The study is carried with care (some technical remarks are appended to the end of the referral), its conclusions are significant and supported by experimental and theoretical evidence. However, I am not convinced that the level of novelty and impact are sufficient to warrant publication in Nat. Commun. To begin with, the results are undoubtedly novel, but not unprecedented. For example, phosphorescence of similar time constants (the afterglow is probably less intense but decays even slower) has been reported in Mn²⁺ doped AlN [H. Zhang, ECS J. Solid State Sci. Technol. 2, R117 (2013)]. Even the attribution of the afterglow to the nitrid vacancies has been made before for both intrinsic and doped AlN [L. Trinkler, Rad. Measurements, 71, 232 (2014), N. J. Cherepy, Opt. Mater. 54, 14 (2016)]. The authors claim that they achieved an intrinsic phosphorescence but to me this is not a significant advantage because they still need a large concentration of crystal defects (vacancies). They also claim that their material do not suffer from the degradation over time, but they do not provide any evidence to support this statement (and for Mn:AlN the degradation seems not to be an issue as well). Finally, the fact that the slow component of the phosphorescence is triggered by X-rays while UV results into faster radiative decay is a remarkable discovery. For the proposed applications in X-ray dosimetry it is not clear to me, though, what are the benefits of AlN over existing materials. Therefore, I believe that the manuscript is more suitable e.g. for Commun. Phys. or Commun. Chem.

Some technical comments

The authors shall include some references to widely studied phosphorescence of AlN and discuss them in the introductory part.

I. 55: The term parity-forbidden excited state (PFES) might be confusing. It is the transition that is

parity forbidden, not the state. In most of the manuscript, PFES refers to the recombination transition (from the PFES to the valence band). However, in l. 56 it is discussed in relation with the excitation pathway. The excitation might include a core s-orbital, though, and be parity allowed.

l. 56: the condition $\Delta k \neq 0$ shall be explained in more detail to facilitate the comprehension

l. 67-71: The statement is unclear and probably misleading. The defect states contain a significant weight of Al-s and N-s orbitals, making the transition from the defect states to the valence band (N-p orbitals) parity-allowed (it can be still prohibited due to other reasons than parity, though).

The same objection holds for l. 183-186.

l. 133: Adding the radiative lifetimes makes no sense.

l. 176: A verb is missing (To further _ the observation)

Language editing is required.

Point by Point Response to the Reviewers

Reviewer: 1 Comments to the Author

The manuscript presents an interesting study of ultralong intrinsic phosphorescence (UIP) in aluminum nitride (AlN) single-crystal scintillators, and proposes a theoretical model which uses the thermal-assisted radiative recombination of parity-forbidden excited state (PFES) electrons. The 1-inch high-quality AlN single crystals notoriously difficult to produce at large sizes are first prepared by the authors. Interestingly, such AlN single crystals exhibit an UIP under X-ray radiation, and the PFES electron recombination is suggested to explain UIP mechanism, which is remarkable and important. Overall, I think that this manuscript merits publication in Nature Communication. Addressing the following minor issues will increase this paper's impact on the readership.

We would like to thank you for reviewing our paper; we appreciate your insightful comments on our research. We have revised the manuscript according to your suggestions and believe that the revisions have improved the paper.

Please find below our detailed responses (in blue) to each comment (in black) provided by the reviewer. In addition, revisions to the original article are highlighted in red here.

1) In Figure 2c, the UIP in AlN single crystal is remarkable, but the data and words in the inset are too small to read, and the fitting results are not detailed enough.

ANS: Thanks for your suggestion. We have adjusted Fig. 2c and marked the decay time obtained by fitting. The details of fitting have been added to the discussion in our revised manuscript as follows :

Page 6:

“The PL decay curves can be fitted by tail functions of different orders to obtain fast and slow components which usually represent different radiation transition pathways. This technique has been widely used in time-resolved fluorescence spectroscopy to study the thermal dynamics of carriers.”

2) “In the AlN SCs, the presence of V_N leads to the UV (352 nm) and yellow (607 nm) emission, and the ultralong (>20000 s) UV phosphorescence originates from the radiative transition of PFES electrons through thermal-assisted process, which is brought by the unique V_N -related electronic structure of AlN.” The authors’ claim that the existence of intrinsic N vacancies in AlN is the major reason of UIP emission should be explained more clearly on its mechanism.

ANS: Thanks for your suggestion. According to the theoretical calculation, near the conduction band minimum (CBM) and valence band maximum (VBM), the nitrogen vacancy defects respectively introduce defect energy levels (Fig. 3 b Fig S3) which usually play as important recombination centers in a phosphorescence emission. Different from conventional phosphor, the electronic structure of AlN phosphor with nitrogen vacancy shows that N 2p orbits contribute to an important component of conduction band, valence band and even defect energy levels, which leads to that the direct transition of excited-state electrons (in N 2p orbit) is forbidden due to parity selection rule. However, through phonon-assisted process, the radiative transition for emitting long phosphorescence can be realized by such excited electrons. To verify the phenomenon that the long phosphorescence in AlN results from nitrogen vacancy, the high-temperature N_2 -riched annealing as a surface treatment of crystal is used to reduce the density of nitrogen vacancy (verified by XPS). The PL spectrum achieved shows that the luminous intensity of N_2 -rich treated AlN obviously decreased, which demonstrates the long phosphorescence practically comes from N vacancy. Therefore, our conclusion has been verified both in theory and experiment.

3) In Figure 3a, the PL spectroscopy of AlN both under X-ray and 266 nm laser can be fitted to three peaks, of which the characteristics of these fitting peaks should be given.

ANS: Thank you for your suggestion. In Fig. 3a, the PL spectrum of AlN can be fitted with Gaussian peak function to obtain three peaks. Under X-ray excitation, the fitted PL peaks are respectively located at 352nm, 607nm and 687nm, whose corresponding photon energies are approximately equal to the energy between conduction band and N vacancy defect energy level.

4) The language and grammar of this manuscript should be more specialized for readability, and more relevant citations should be added.

ANS: Thank you for your suggestion. We have asked professional translator of scientific English to polish the manuscript for readability, and more suitable references have been added to the revised version.

[22] J. Xu, N. J. Cherepy, J. Ueda & S. Tanabe. Red persistent luminescence in rare earth-free AlN:Mn²⁺ phosphor. *Materials Letters* 206, 175-177 (2017).

[23] I. A. Weinstein, A. S. Vokhmintsev & D. M. Spiridonov. Thermoluminescence kinetics of oxygen-related centers in AlN single crystals. *Diamond and Related Materials* 25, 59-62 (2012).

[24] A. T. Wieg, E. H. Penilla, C. L. Hardin, Y. Kodera & J. E. Garay. Broadband white light emission from Ce:AlN ceramics: High thermal conductivity down-converters for LED and laser-driven solid state lighting. *APL Materials* 4, 126105 (2016).

[25] L. Trinkler & B. Berzina. Recombination luminescence in aluminum nitride ceramics. *physica status solidi (b)* 251, 542-548 (2014).

Reviewer: 2 Comments to the Author

The manuscript reports ultralong phosphorescence from AlN with a high concentration of N vacancies. The study is carried with care (some technical remarks are appended to the end of the referral), its conclusions are significant and supported by experimental and theoretical evidence.

However, I am not convinced that the level of novelty and impact are sufficient to warrant publication in Nat. Commun. To begin with, the results are undoubtedly novel, but not unprecedented. For example, phosphorescence of similar time constants (the afterglow is probably less intense but decays even slower) has been reported in Mn²⁺-doped AlN [H. Zhang, ECS J. Solid State Sci. Technol. 2, R117 (2013)]. Even the attribution of the afterglow to the nitride vacancies has been made before for both intrinsic and doped AlN [L. Trinkler, Rad. Measurements, 71, 232 (2014), N. J. Cherepy, Opt. Mater. 54, 14 (2016)]. The authors claim that they achieved an intrinsic phosphorescence but to me this is not a significant advantage because they still need a large concentration of crystal defects (vacancies). They also claim that their material do not suffer from the degradation over time, but they do not provide any evidence to support this statement (and for Mn:AlN the degradation seems not to be an issue as well). Finally, the fact that the slow component of the phosphorescence is triggered by X-rays while UV results into faster radiative decay is a remarkable discovery. For the proposed applications in X-ray dosimetry it is not clear to me, though, what are the benefits of AlN over existing materials.

Therefore, I believe that the manuscript is more suitable e.g. for Commun. Phys. or Commun. Chem.

Thank you for your review and the positive comments on our research. We believe the novelty and significance of our work are sufficient based on the following parts.

First, we presented, for the first time, a record-breaking ultra-long phosphorescence (more than 20000s) in AlN single crystal under X-ray excitation. The ultra-long phosphorescence was produced without the need of doping additional rare-earth metals, which is different from the works reported previously [H. Zhang, ECS J. Solid State Sci. Technol. 2, R117 (2013)]. Second, we prepared a large-size AlN single crystal (up to 1 inch), which is a remarkable advance for AlN material. Compared with AlN ceramic materials (L. Trinkler, Rad. Measurements, 71, 232 (2014)), this single crystal obviously has much lower density of defects. It has been recognized that intrinsic phosphorescence is the primary origin of native (undoped) defects. However, our research interest is on the ultralong intrinsic phosphorescence in AlN single crystal (from X-ray excitation rather than UV) and its close relation to N vacancy (verified by theoretical calculation). We believe you have agreed that this finding is a “remarkable discovery” based on your comment.

For X-ray dosimetry, we can refer to AlN's application to ultraviolet dosimetry [L. Trinkler, et al, Nuclear Instruments and Methods in Physics Research Section A: Accelerators, Spectrometers, Detectors and Associated Equipment 2007, 580, 354.]. As is known, X-ray has more extensive and important applications than UV at present. In addition, the current X-ray dosimetry applications are mostly based on

thermoluminescence of materials, that is, the energy of incident photons is converted to heat, then the heat is converted to persistent luminescence. In our work, we put forward a mechanism by using the photon-induced phosphorescence in AlN to achieve higher efficiency and accuracy, which may provide a new pathway for researches and applications of high-energy radiation dosimetry.

In conclusion, we believe that our work does have enough significance and progress. Besides, based on your suggestion, this manuscript has been revised and improved a lot.

Please find below our detailed responses (in blue) to each comment (in black) provided by the reviewer. In addition, revisions to the original article are highlighted in red here.

Some technical comments

1) The authors shall include some references to widely studied phosphorescence of AlN and discuss them in the introductory part.

ANS: Thanks for your suggestion. We have read more literature on AlN phosphorescence with a further understanding of research background. Relevant contents have been added to the introduction section in the revised manuscript.

Page 3:

“Aluminum nitride (AlN) is well known as a inorganic semiconductor with an ultrawide bandgap of about 6.2 eV, and its properties of phosphorescence, photoluminescence (PL) and thermoluminescence (TL) have attracted much attention.²²⁻²⁷ Here, we presented an ultraviolet (UV) ultralong intrinsic phosphorescence (UIP, > 20,000 s) of AlN single crystal under X-ray excitation.”

[22] J. Xu, N. J. Cherepy, J. Ueda & S. Tanabe. Red persistent luminescence in rare earth-free AlN: Mn²⁺ phosphor. *Materials Letters* 206, 175-177 (2017).

[23] I. A. Weinstein, A. S. Vokhmintsev & D. M. Spiridonov. Thermoluminescence kinetics of oxygen-related centers in AlN single crystals. *Diamond and Related Materials* 25, 59-62 (2012).

[24] A. T. Wieg, E. H. Penilla, C. L. Hardin, Y. Kodera & J. E. Garay. Broadband white light emission from Ce:AlN ceramics: High thermal conductivity down-converters for LED and laser-driven solid state lighting. *APL Materials* 4, 126105 (2016).

[25] L. Trinkler & B. Berzina. Recombination luminescence in aluminum nitride ceramics. *physica status solidi (b)* 251, 542-548 (2014).

2) l. 55: The term parity-forbidden excited state (PFES) might be confusing. It is the transition that is parity forbidden, not the state. In most of the manuscript, PFES refers to the recombination transition (from the PFES to the valence band). However, in l. 56 it is discussed in relation with the excitation pathway. The excitation might include a

core s-orbital, though, and be parity allowed.

ANS: Thank you for your suggestion. The “PFES” is used to describe the electrons in unsteady state (the electrons whose direct transitions are forbidden due to parity selection rule in excited states). For the description on “PFES” at line 56, we admit that it is not appropriate. Modifications to this description have been shown in the revised manuscript.

3) l. 56: the condition $\Delta\mathbf{K}\neq\mathbf{0}$, shall be explained in more detail to facilitate the comprehension

ANS: Thank you for your suggestion. In general, the direct transition between conduction band minimum (CBM) and valence band maximum (VBM) is allowed in the origin point of wave vector space ($\mathbf{K}=\mathbf{0}$). However, for several semiconductors, the quantum selection rule (parity selection rule) states that direct transitions at $\mathbf{K}=\mathbf{0}$ are forbidden, that is, the probability (W) of direct transitions is zero at $\mathbf{K}=\mathbf{0}$. However, considering quadratic term perturbation Hamiltonian (a. Smith, R Wave Mechanics of Crystalline Solids. 2ed Ed. Chapman and Hall), even though $\mathbf{K}=\mathbf{0}$ and $W=0$, the transition probability is not equal to zero when $\Delta\mathbf{K}\neq\mathbf{0}$. Such transition often needs phonons' assistance. Relevant contents have been added to the revised version.

Page 9:

“The excited-state electrons can be excited by external excitation (such as phonon excitation, $\Delta\mathbf{K}\neq\mathbf{0}$), and then result in radiative transitions with persistent phosphorescence.^{2,21} In general, the direct transition between conduction band minimum (CBM) and valence band maximum (VBM) is allowed in the origin point of wave vector space ($\mathbf{K}=\mathbf{0}$). However, for several semiconductors, the quantum selection rule (parity selection rule) states that direct transitions at $\mathbf{K}=\mathbf{0}$ are forbidden, that is, the probability (W) of direct transitions is zero at $\mathbf{K}=\mathbf{0}$. However, considering quadratic term perturbation Hamiltonian, even though $\mathbf{K}=\mathbf{0}$ and $W=0$, the transition probability is not equal to zero when $\Delta\mathbf{K}\neq\mathbf{0}$. Such transition often needs phonons' assistance.²¹”

[21] R. A. Smith & E. T. Jaynes. Wave Mechanics of Crystalline Solids. American Journal of Physics 30, 846-847 (1962).

4) l. 67-71: The statement is unclear and probably misleading. The defect states contain a significant weight of Al-s and N-s orbitals, making the transition from the defect states to the valence band (N-p orbitals) parity-allowed (it can be still prohibited due to other reasons than parity, though). The same objection holds for l. 183-186.

ANS: Thanks for your suggestion. We have noticed the inaccuracy of this description, so more accurate and clearer depiction of the fluorescence mechanism has been added here.

Page 3:

“However, due to quantum selection rule, only few excited electrons (those satisfy both the momentum and parity conditions) can obtain fast fluorescence through direct radiative transitions, and most of them will obtain persistent phosphorescence through phonon-assisted transitions and in-band multi-process transitions. The process above is verified by observing the V_N -related electronic structure configuration where the N 2p orbits have significant weights of conduction band, valence band and even defect energy levels.”

Page 8:

“For V_N -introduced energy band structure and partial density of states (PDOS), N 2p orbits exert valence band, conduction band and even defect energy level simultaneously (Fig. 3c), but the direct transitions between N 2p orbits will be forbidden because of the parity selection rule. Thus, only few excited-state electrons (those satisfy both the momentum and parity conditions) can achieve direct transitions to obtain a fast fluorescence, and most of them are still in an unstable excited state for a long time, similar to forming store of energy in phosphor.”

5) l. 133: Adding the radiative lifetimes makes no sense.

ANS: Thank you for your comment. The PL decay curve can be fitted by the tail functions with different orders to obtain fast and slow components which usually represent different radiation transition pathways. This technique is widely used in time-resolved fluorescence spectroscopic analysis for the study of carriers' thermodynamic.

6) l. 176: A verb is missing (To further _ the observation)

ANS: Thank you for suggestion. We have added a verb here.

“To further **understand** the observation on the UIP mechanism of AlN SCs”

Language editing is required.

ANS: Thank you for your suggestion. We have asked professional translator of technical English to polish the manuscript for readability.

Reviewer #1 (Remarks to the Author):

All my previous concerns are well addressed.

Reviewer #2 (Remarks to the Author):

In their response letter, the authors have provided a reasonable explanation of the novelty and significance of the manuscript. The manuscript reports several achievements (large single crystals of AIN, long and intrinsic phosphorescence), each of them itself being rather significant. Together, they represent significance and novelty that I consider boundary for acceptance to Nat. Commun. Further, the authors studied their subject very thoroughly and I have read the manuscript with pleasure. Therefore, I recommend to accept the manuscript.

As for the technical comments, I still insist that the exponential fits presented in Fig. 2c and related text are not correct. This is not detrimental to the message of the manuscript, but a correction is necessary prior to acceptance. I was probably too brief in my previous report since I believed that the mistake is obvious. I will provide more details now.

I suppose that the red curve in Fig. 2c was fit with function $I(t) = I_0 \exp(-t/\tau)$. From the graph I can estimate two points (first and third red symbol): $t_0=0$, $I_0 \sim 100$ and $t \sim 1200$ s, $I \sim 15$. From them, I estimate $\tau = t / \log(I_0/I) \sim 600$ s. The value 5.66 s reported by the authors is clearly incorrect (possibly due to misplacement of decimal point). The same comment holds also for the black curve.

I suppose that the black curve in Fig. 2c was fit with function $I(t) = I_1 \exp(-t/\tau_1) + I_2 \exp(-t/\tau_2)$. It corresponds to two populations taking two different recombination paths, fast and slow. In turn, two time constants are obtained, τ_1 and τ_2 . The authors have added the two time constants up, 10.60 s + 101.59 s = 112.19 s. However, it is absolutely unjustified to do this and the resulting time of 112.19 s has no physical meaning. One can perhaps add the time constants of two sequential processes to obtain (approximate) time constant of the composed process, but this is certainly not applicable to two parallel processes. Also note that the reported decrease of overall intensity from 100 % in 0 s to 24 % in 3600 s corresponds to $\tau = 3600 / \log(100/24) \sim 2500$ s, while for the decrease to 15 % in 7200 s $\tau \sim 7700$ s because the slow process becomes more important. This illustrates the fact that the single time constant cannot be used.

As for the language, the authors shall unify the abbreviations for fluorescence (Flour., Fluor.) to Fluor. and the abbreviations for phosphorescence (Phosp., Phorp.) to Phosph. (ph. represents single Greek letter and shall not be split).

Point by Point Response to the Reviewers

Reviewer: 2 Comments to the Author

In their response letter, the authors have provided a reasonable explanation of the novelty and significance of the manuscript. The manuscript reports several achievements (large single crystals of AlN, long and intrinsic phosphorescence), each of them itself being rather significant. Together, they represent significance and novelty that I consider boundary for acceptance to Nat. Commun. Further, the authors studied their subject very thoroughly and I have read the manuscript with pleasure. Therefore, I recommend to accept the manuscript.

Thank you for your review and approval to this paper's publication. Our manuscript has been improved after revision according to your valuable and insightful suggestion.

Please find below our detailed responses (in blue) to each comment (in black) provided by the reviewer. In addition, revisions to the original article are highlighted in red here.

1) As for the technical comments, I still insist that the exponential fits presented in Fig. 2c and related text are not correct. This is not detrimental to the message of the manuscript, but a correction is necessary prior to acceptance. I was probably too brief in my previous report since I believed that the mistake is obvious. I will provide more details now. I suppose that the red curve in Fig. 2c was fit with function $I(t)=I_0\exp(-t/\tau)$. From the graph I can estimate two points (first and third red symbol): $t_0=0$, $I_0\sim 100$ and $t\sim 1200$ s, $I\sim 15$. From them, I estimate $\tau=t/\log(I_0/I)\sim 600$ s. The value 5.66 s reported by the authors is clearly incorrect (possibly due to misplacement of decimal point). The same comment holds also for the black curve. I suppose that the black curve in Fig. 2c was fit with function $I(t)=I_1\exp(-t/\tau_1)+I_2\exp(-t/\tau_2)$. It corresponds to two populations taking two different recombination paths, fast and slow. In turn, two-time constants are obtained, τ_1 and τ_2 . The authors have added the two-time constants up, 10.60 s + 101.59 s = 112.19 s. However, it is absolutely unjustified to do this and the resulting time of 112.19 s has no physical meaning. One can perhaps add the time constants of two sequential processes to obtain (approximate) time constant of the composed process, but this is certainly not applicable to two parallel processes. Also note that the reported decrease of overall intensity from 100 % in 0 s to 24 % in 3600 s corresponds to $\tau=3600/\log(100/24)\sim 2500$ s, while for the decrease to 15 % in 7200 s $\tau\sim 3700$ s because the slow process becomes more important. This illustrates the fact that the single time constant cannot be used.

ANS: First of all, we really appreciate your question and your insightful opinions on this issue. In our previous manuscript, to study the radiative transition paths in AlN

SCs, we used a two-order decay function to fit the decay curve of phosphorescence through the fitting time constant (τ , usually is a mean time).

According to your comments, we realized that it's unjustified to add up the two fitting time constants (τ_1 and τ_2). In addition, such time constants in Fig. 2c may mislead readers into taking it as the persistent time of phosphorescence rather than the lifetime of radiation transitions in practice. Thus, to quantitatively characterize the phosphorescence, a persistent time (T_d) is defined as the time interval during which the emitting intensity decreases from 100% to 10%. The persistent time of UV and yellow phosphorescence are calculated to be 9360 s and 4320 s, respectively. Relevant and detailed discussions have been added to the revised manuscript as follows:

Page 6:

“In Fig. 2c, both of the decay curve of UV and yellow phosphorescence can be well fitted by the two-order decay function, from which a fast time constant ($\tau_1 \sim 5$ s) and a slow time constant ($\tau_2 \sim 110$ s) can be extracted. The different time constants indicate that there are two different radiation recombination paths in AlN SCs, and thus an ultralong phosphorescence is produced. To quantitatively characterize the phosphorescence, a persistent time (T_d) is defined as the time interval during which the emitting intensity decreases from 100% to 10%. Calculation shows that the persistent time of UV and yellow phosphorescence in AlN SCs are 9360 s and 4320 s, respectively, from which it can be obviously concluded that the UV phosphorescence shows a longer persistent time than that of yellow phosphorescence, indicating a more significant weight of slow radiation recombination component in UV phosphorescence.”

2) As for the language, the authors shall unify the abbreviations for fluorescence (Flour., Fluor.) to Fluor. and the abbreviations for phosphorescence (Phosp., Phorp.) to Phosph. (ph. represents single Greek letter and shall not be split).

ANS: Thank you for your suggestion. Relevant parts have been revised in the

manuscript.

Reviewer #2 (Remarks to the Author):

Unfortunately, the last revision was only partially satisfactory. I am still puzzled with Fig. 2c. Persistent times added on the last revision correspond well to the presented data. However, the values of τ_1 and τ_2 are not consistent. There is clearly some misunderstanding that needs to be resolved. I would like to see the formula of the function fitted to the data shown in Fig. 2c. So far, I cannot imagine any form of the second-order exponential function that would fit the data with the values of τ equal to 5 s and 110 s. For single exponential function it takes about $2 \cdot \tau$ [$\log(10) \cdot \tau$] for the intensity to drop to 10 % of the original value and for the second-order exponential it shall take between $2 \cdot \tau_1$ and $2 \cdot \tau_2$.

In addition, the caption of Fig. 2c reports somewhat different times τ_1 , τ_2 than the text in line 127.

Point by Point Response to the Reviewers

Reviewer: 2

Unfortunately, the last revision was only partially satisfactory. I am still puzzled with Fig. 2c. Persistent times added on the last revision correspond well to the presented data. However, the values of τ_1 and τ_2 are not consistent. There is clearly some misunderstanding that needs to be resolved. I would like to see the formula of the function fitted to the data shown in Fig. 2c. So far, I cannot imagine any form of the second-order exponential function that would fit the data with the values of τ equal to 5 s and 110 s. For single exponential function it takes about $2 \cdot \tau$ [$\log(10) \cdot \tau$] for the intensity to drop to 10 % of the original value and for the second-order exponential it shall take between $2 \cdot \tau_1$ and $2 \cdot \tau_2$.

In addition, the caption of Fig. 2c reports somewhat different times τ_1 , τ_2 than the text in line 127.

ANS: Thank you for your profound comment on the issue of Fig. 2c. In Fig. 2c, the decay curve of phosphorescence is well fitted to bi-exponential decay function (Fig. R1), but poorly fitted to mono-exponential decay function (Fig. R2), which means that excited carriers produce the phosphorescence through two radiation recombination paths. For a clearer detail, the fitting results and the raw data are presented here.

In response to your question, we found that the unit of fitting τ_1 and τ_2 in our last reply was in minutes (see Fig. R3) rather than seconds. We are sorry for the confusion caused by this mistake and we have got it corrected.

At last, we greatly appreciate your insightful and responsible attitude in reviewing our manuscript. Thank you sincerely for your advice.

Figure R1. The fitting results by bi-exponential decay function.

Figure R2. The fitting results by mono-exponential decay function.

Figure R3. The fitting results in last revised manuscript.

Raw data:

Time (s)	Time (mins)	UV Intensity (a.u.)	Yellow Intensity(a.u.)
0	0	207.01096	100.57526
720	12	102.77626	18.46228
1440	24	60.90945	13.72932
2160	36	51.57833	11.37033
2880	48	42.49638	11.62524
3600	60	39.19037	10.67634
4320	72	34.53265	10.07679
5040	84	31.88568	9.90134
5760	96	28.77339	8.57404
6480	108	27.76648	9.69538
7200	120	25.66111	7.46034
7920	132	24.3033	10.70993
8640	144	23.22773	9.1843
9360	156	20.77909	8.25367
10080	168	19.92473	8.8868
10800	180	19.29922	8.1011
11520	192	19.45941	8.15449
12240	204	17.35405	7.10944
13680	228	16.9345	7.30777
14400	240	15.05798	6.93399
15120	252	15.72925	6.50682
15840	264	14.2799	7.03316
16560	276	14.47823	6.55258
17280	288	14.31042	7.22386
18000	300	14.89778	8.1545
18720	312	13.28824	6.30085
19440	324	12.5712	6.40765
20160	336	12.51017	6.91873

Reviewer #2 (Remarks to the Author):

With the last revision, the issue with the time constants has been resolved. The paper can be published in its present form.

Point by Point Response to the Reviewers

Reviewer 2:

With the last revision, the issue with the time constants has been resolved. The paper can be published in its present form.

ANS: Thank you for reviewing our manuscript.